# The Role of BCL-2 Expression in Patients with Myelodysplastic Neoplasms

**DOI:** 10.3390/cimb47050346

**Published:** 2025-05-10

**Authors:** Bartłomiej Kuszczak, Krzysztof Zduniak, Angela Jendzierowska, Tomasz Wróbel, Piotr Ziółkowski, Justyna Rybka

**Affiliations:** 1Department and Clinic of Hematology, Cellular Therapies and Internal Medicine, Medical University of Wrocław, 50-367 Wroclaw, Poland; bartkuszczak@gmail.com (B.K.);; 2Department of Pathology, Lower Silesian Center of Oncology Pulmonology and Hematology, 53-439 Wroclaw, Poland; 3Department of Clinical and Experimental Pathology, Faculty of Medicine, Medical University of Wrocław, 50-367 Wroclaw, Poland; piotr.ziolkowski@umw.edu.pl

**Keywords:** MDS, BCL-2, trephine biopsy, stem cells

## Abstract

Myelodysplastic neoplasms (MDS) represent a heterogeneous group of neoplastic bone marrow disorders. A crucial component in regulating bone marrow cell apoptosis is the B-cell CLL/lymphoma 2 (BCL-2) protein. This retrospective study aimed to assess BCL-2 expression by immunohistochemistry in trephine biopsy specimens from 76 patients diagnosed with MDS. The obtained retrospective results were correlated with clinical parameters, including age, sex, MDS subtype, IPSS, IPSS-R, bone marrow blast percentage, Ogata score, response to treatment, blood morphology parameters, and overall survival (OS). The median follow-up duration was 16 months. During the observation period, 58 patients died (median OS of this group: 14.6 months), and 25 patients experienced progression to acute myeloid leukemia. The median BCL-2 expression assessed using the Histoscore (H-score) was 10. Patients with BCL-2 expression below 10 had better survival outcomes than those with expression ≥ 10. Furthermore, patients without detectable BCL-2 expression had significantly better survival compared to those with detectable BCL-2 expression (*p* = 0.0084). Higher BCL-2 expression was significantly associated with high and very high cytogenetic risk, as defined by IPSS-R. BCL-2 immunohistochemistry should be viewed as a complementary biomarker that, when integrated with IPSS-R and mutational data, could refine therapeutic algorithms.

## 1. Introduction

Myelodysplastic syndromes (MDS) encompass a heterogeneous group of bone marrow neoplasms. Despite their varied clinical presentation and course, the hallmark features of MDS include the presence of dysplasia, at least one lineage cytopenia in the peripheral blood, and the potential for progression to acute myeloid leukemia (AML) [1]. MDS are classified as a rare diseases, with an incidence of 4 per 100,000 annually. They are rarely diagnosed before age 50 (<10% of cases), and the incidence increases with age (median age: 76 years). Although known risk factors exist (e.g., chemotherapy, smoking, obesity), the cause remains unknown in most cases [2,3,4]. This is attributable to the complex pathogenesis driven by the accumulation of random point mutations in the genome [1]. Increasing attention has been directed at immune dysregulation and the bone marrow niche microenvironment [5,6]. BCL-2, a key member of the BCL-2 protein family, regulates apoptosis by interacting with pro-apoptotic proteins, such as BAX and BAK, to inhibit intrinsic apoptotic pathways [7]. Elevated BCL-2 expression has been reported in various solid tumors (e.g., breast, lung, esophageal, and prostate cancers), non-Hodgkin B-cell lymphomas, and MDS. The literature highlight that there is elevated BCL-2 expression in early myeloid precursors among patients with advanced MDS stages and AML progression. Paradoxically, some low-risk MDS patients exhibited lower overall survival (OS) despite favorable prognostic scores (e.g., IPSS), potentially due to enhanced BCL-2 activity [8,9,10]. The efficacy of BCL-2 inhibitors in treating other myeloid malignancies has led to the exploration of similar molecular mechanisms for MDS therapy [11,12,13]. Despite advances in understanding MDS pathogenesis and new therapeutic agents, prognoses remain poor. The median OS of high-risk MDS patients treated with azacitidine (AZA) was 17.5 months, with complete remission (CR) achieved in only 10–32% of patients. Poor response to hypomethylating agents (HMAs) was associated with an OS of <6 months [11]. According to the European Society for Medical Oncology (ESMO) guidelines, histopathological bone marrow examination is a standard practice in nearly all MDS cases [14]. This study aimed to evaluate the prognostic significance of BCL-2 expression in trephine biopsies and its potential utility in guiding therapeutic decisions for MDS patients.

## 2. Materials and Methods

This retrospective study analyzed trephine biopsy specimens from 104 patients who were treated at the Department of Hematology, Cellular Therapies, and Internal Medicine, University Clinical Hospital in Wroclaw, between 2015 and 2021. Twenty-eight specimens were excluded due to incomplete medical data, suboptimal specimen quality, or unconfirmed MDS diagnosis. Diagnostic specimens were obtained at the initial diagnosis, excluding follow-up biopsy specimens. The final cohort comprised 76 patients with confirmed MDS diagnoses, including 43 males and 33 females. The median age was 66 years (range: 23–91 years). MDS subtypes were classified using the 2022 World Health Organization (WHO) criteria: MDS IB-2 (33 patients), MDS LB (24), MDS IB-1 (11), MDS-F (6), and one each of the hypoplastic and del(5q) MDS subtypes. Treatment regimens included subcutaneous AZA (48 patients, median of six cycles), intensive chemotherapy (six patients), lenalidomide (five patients), and other therapies (two patients) (Table 1). There were 15 patients ineligible for systemic therapy, and 21 patients who underwent allogeneic hematopoietic stem cell transplantation. Ethical approval was obtained from the Bioethics Committee of Wroclaw Medical University (approval number: KB—753/2020).

### 2.1. Specimen Preparation

Trephine biopsy specimens were fixed in 10% formalin, decalcified using Oxford reagent, and embedded in paraffin. Four-micron-thick sections were stained immunohistochemically using the Autostainer Link 48 system, primary anti-BCL-2 antibody (clone 124, Agilent, catalog number IR61461-2), and the Envision FLEX+ visualization system (Agilent, catalog number K800221-5) (Perlan Technologies Polska Warszawa).

### 2.2. Staining Technique Evaluation

Immunohistochemical staining was evaluated using the Histoscore (H-score) method. Reactive lymphocytes in bone marrow served as internal controls. Myeloid lineage staining intensity was graded from 0 (no reaction) to 3 (strong reaction comparable to reactive lymphocytes). For each intensity level, the percentage of stained cells was recorded, and the H-score was calculated using the formula: H-score = intensity 1 × percentage of cells + intensity 2 × percentage of cells + intensity 3 × percentage of cells. The possible score range was 0–300 (Figure 1).

### 2.3. Statistical Analysis

Statistical analyses were performed using the Real Statistics Resource Pack for Microsoft Excel (version 15.0.5023.1000, Microsoft, Redmond, WA, USA) and GraphPad Prism (version 8.0.1, GraphPad Software, La Jolla, CA, USA). BCL-2 expression levels were correlated with clinical and laboratory parameters, including age, sex, MDS subtype, cytogenetic risk, IPSS, IPSS-R, blast percentage, Ogata score, response to treatment, blood morphology parameters [hemoglobin, white blood cells (WBCs), neutrophils (ANCs), lymphocytes (LYMPs), monocytes (AMCs), PLT], and OS. Statistical tests included the Mann–Whitney U test (sex and infection associations), Spearman’s rank correlation coefficient (age and blast percentage), and the Kruskal–Wallis test (MDS subtype, cytogenetic risk, IPSS-R, IPSS, and treatment response). Kaplan–Meier curves and log-rank tests were used for survival analysis. A *p*-value < 0.05 was considered statistically significant. Patients were grouped based on BCL-2 expression for survival analysis using statistical parameters, such as mean, median, and quartiles.

## 3. Results

Apart from reactive lymphocytes and plasma cells, staining was observed in the myeloid lineage. The reaction was primarily noted in immature cells of the granulocytic series, including myeloblasts, promyelocytes, and myelocytes.

This cohort comprises 76 MDS patients aged 23–91 years (median 66) who uniformly exhibit cytopenias, as evidenced by a median hemoglobin of 9.3 g/dL, low leukocyte (2.76 × 10^3^/µm) and neutrophil (1.28 × 10^3^/µm) counts, and reduced platelets (PLTs) (81 × 10^3^/µm) (Table 1). The distribution of MDS subtypes is heterogeneous, with common forms like MDS-IB2 (33 patients) and MDS-LB (24 patients) being more prevalent, while rarer entities such as MDS with isolated 5q deletion and hypoplastic MDS are represented by only a single case each. Similarly, risk stratification reveals that a significant portion of the cohort falls into the high or very high IPSS-R risk groups (22 and 21 patients, respectively). Of the 25 MDS patients who evolved to AML, 3 belonged to the “low” category, 4 to the “intermediate” group, and 18 to the “high” group (merging high/very high); due to the limited numbers in the low and intermediate groups, no clear differences in BCL-2 expression emerged among these categories.

### 3.1. BCL-2 Expression and Survival

The median follow-up period was 16 months. During this time, 58 patients died (median OS: 14.6 months), and 25 progressed to AML. The mean BCL-2 expression was 23.88, with a median of 10 (range: 0–140). There was no statistically significant difference in survival between patients with high and low BCL-2 expression (*p* = 0.6272). After isolating the population with BCL-2 expression < 10, it was observed that this group correlated with longer survival (median OS = 26.57 months) compared to patients with BCL-2 expression ≥ 10 (median OS 13.90 months) (*p* = 0.0341) (Figure 1). The median H-scores for BCL-2 in the different risk groups and MDS subtypes are presented in Table 2.

Patients in the first quartile had an H-score of 0 (no expression). It was shown that patients without BCL-2 expression had significantly longer survival compared to those with detectable BCL-2 expression (*p* = 0.0084) (Figure 2).

This correlation was not observed in the third quartile, where all subjects had an H-score of 35 (*p* = 0.9082). These patients did not differ in survival compared to those with a BCL-2 expression > 10.

### 3.2. BCL-2 Expression and Cytogenetic Risk

No statistically significant differences in BCL-2 expression were observed among patients with varying cytogenetic risk levels defined by IPSS-R cytogenetic risk [15] (*p* = 0.1585, Kruskal–Wallis test). However, after grouping patients into two categories based on IPSS-R cytogenetic risk (very low, low, intermediate [0–2 points] vs. high, very high [3–4 points]), it was noted that BCL-2 expression was higher in patients with high and very high risk (median H-score: 40; interquartile range [IQR]: 20–75) compared to those with very low to intermediate risk (median H-score: 10; IQR: 0–35, *p* = 0.0496, Mann–Whitney U test).

### 3.3. BCL-2 Expression and IPSS-R

No statistically significant differences in BCL-2 expression were noted between patients in different IPSS-R risk groups (very low/low, intermediate, high, very high) (*p* = 0.0854, Kruskal–Wallis test). However, when dividing patients into two groups based on IPSS-R risk categories (very low, low, intermediate, high [0–3 points] vs. very high [4 points]), the H-score was significantly higher in the very high-risk group (median H-score: 45; interquartile range [IQR]: 20–75) compared to the lower-risk group (median H-score: 20; IQR: 0–40) (*p* = 0.0130, Mann–Whitney U test).

No association was found between H-score and age, sex, type of MDS (according to WHO 2022), IPSS, occurrence of infections, or treatment response.

## 4. Discussion

The BCL-2 protein is one of the many molecules involved in the pathogenesis of MDS. Evidence suggests that a disruption in the balance of BCL-2 family proteins, favoring anti-apoptotic proteins (including BCL-2), is associated with shorter survival, progression to more advanced stages, and adverse prognostic factors [9,10,16,17]. The findings of this study align with previous observations, particularly regarding the correlation between the intensity of BCL-2 expression and patient OS. However, this study did not demonstrate a correlation between BCL-2 levels and MDS subtype (according to WHO 2022) or blast count, which has been reported in the available literature. These discrepancies may stem from differences in the methodology used by the authors. Most prior studies were based on an immunophenotypic assessment of BCL-2 levels in cells rather than on immunohistochemical staining methods [18,19]. Additionally, the specificity of the Polish population and the limited sample size may have influenced the results. Detecting BCL-2 in cells through immunohistochemical staining appears to have prognostic significance, independent of the number of stained blasts or staining intensity (a significant difference was observed between the H-score 0 group and the rest). Even a small population of BCL-2+ cells may eventually dominate in the pathological tumor environment due to greater resistance to apoptosis. The authors agree that the hypothesis that these cells drive disease progression and shorter patient survival requires further investigation.

A noteworthy observation is the higher H-score in patients with high and very high cytogenetic risk. This suggests that BCL-2 may inhibit apoptosis, enabling the survival of cancer cells and facilitating the accumulation of complex and adverse genetic mutations. It is worth mentioning that none of the patients in the studied population were treated with BCL-2 inhibitors. However, the literature includes data from clinical trials evaluating the efficacy and safety of BCL-2 inhibitors, with venetoclax (VEN) being the most commonly studied agent. A Phase 1–2 study (NCT04160052) included 23 patients with high-risk MDS and chronic myelomonocytic leukemia. The patients were treated with a combination of AZA and VEN. The overall response rate (ORR) was 87%, with acceptable toxicity levels. The most common complications were grade 3–4 neutropenia (39% of patients), thrombocytopenia (39%), pneumonia (30%), and three cases of death due to sepsis [20]. Another Phase 1b study by Zeidan et al. included 44 patients with refractory or relapsed MDS who received AZA with VEN. Complete remission (CR) was achieved in 7% of the patients, and marrow CR (mCR) was achieved in 32% of the patients. Additionally, 36% (16/44) of the patients achieved transfusion independence for red blood cells and/or platelets. Among patients with mCR, 43% (6/14) showed hematologic improvement. The median time to achieve CR/mCR was 1.2 months, and the median duration of response for CR + mCR was 8.6 months. The median OS was 12.6 months. The toxicity profile in both studies was similar [11]. In a large 2023 meta-analysis that included 1615 patients diagnosed with MDS or AML, 67.6% of the patients in the MDS subgroup achieved CR or incomplete CR (CRi). The results of a Phase 3 (NCT0440174) trial comparing AZA-VEN with AZA-placebo could prove groundbreaking in establishing a new standard of care [21].

An open question remains whether patients with higher BCL-2 expression respond differently to BCL-2 inhibitors than those with lower BCL-2 expression. A clear answer could help identify the population most likely to benefit from targeted therapy. Conversely, patients predicted to have a poor response to BCL-2 inhibitors may be excluded from such therapy due to potential toxicity.

Given the currently limited treatment options for MDS, BCL-2 inhibitors appear to be one of the most promising therapeutic approaches for improving OS outcomes. Targeted therapy represents the future of hematologic cancer treatment. Our findings suggest the potential benefit of incorporating additional BCL-2 staining during routine diagnostics and trephine biopsy analysis in MDS patients. However, the current analysis is preliminary, and future studies are planned to include a larger population of MDS patients.

Our findings reinforce the concept that dysregulation of anti-apoptotic signaling, especially via BCL-2, can shape the clinical course of MDS and affect survival outcomes. In our cohort, patients whose trephine biopsies displayed no BCL-2 expression (H-score = 0) showed significantly longer overall survival (OS), in line with previous observations that BCL-2 overexpression may confer a proliferation and survival advantage to malignant clones and promote disease progression [22,23]. Notably, we observed that even a small subset of BCL-2+ cells may, over time, expand and become the dominant malignant population, an insight echoed by research illustrating that MDS stem cells enriched in anti-apoptotic pathways can outcompete their healthy counterparts [24].

Although multiple studies have reported an association between BCL-2 expression and higher blast counts or more advanced disease phenotypes, we did not observe a robust correlation between BCL-2 positivity and MDS subtype (WHO 2022). Moreover, we did not detect any clear association with the blast percentage. This may be partly attributable to different detection methodologies used across studies. It should be noted that many prior investigations rely primarily on flow cytometry or RNA-based techniques instead of immunohistochemical staining [25,26]. However, immunohistochemistry, as used in our study, can highlight BCL-2 positivity at the tissue level and detect lower-frequency clonal populations, which may account for subtle differences in reported prevalence and prognostic implications.

Our finding that higher-risk cytogenetic categories (high and very high by IPSS-R) exhibited more pronounced BCL-2 expression suggests that BCL-2 dysregulation could be one of several contributory factors allowing cells with complex or adverse karyotypic aberrations to persist and accumulate [22,26]. Some investigators similarly observed that MDS clones with poorer-risk cytogenetics show an elevated reliance on anti-apoptotic pathways [27]. These overlapping lines of evidence lend biologic plausibility to the notion that targeting BCL-2 in poorer-risk cytogenetic subgroups might bring some therapeutic advantages.

Our analysis also confirms that patients with higher BCL-2 expression had inferior OS, paralleling emerging data that anti-apoptotic regulators such as BCL-2, BCL-XL, and MCL-1 can act as “pro-survival” crutches for MDS cells [23,27]. Several recently published clinical trials involving venetoclax-based combinations (typically with azacitidine or decitabine) have demonstrated promise in high-risk and relapsed or refractory MDS, achieving complete or near-complete remissions in a proportion of patients [23,28]. However, resistance mechanisms—often involving MCL-1 upregulation—have already been observed, underlining the ongoing need for comprehensive biomarker studies and combination approaches [29].

Interestingly, the prognosis for those with BCL-2 expression in the intermediate-range H-score (e.g., above 10 but below the extreme quartiles) did not differ substantially from those at the higher end of expression. It is possible that once BCL-2 is activated beyond a certain biological threshold, additional perturbations in these pathways play a determinative role. The threshold effect mirrors the findings reported by some groups who proposed that minor BCL-2+ subclones can eventually dominate in a conducive tumor microenvironment [26]. In contrast, patients with complete absence of BCL-2 expression may retain intact apoptotic pathways (as reflected in the more favorable OS we observed).

From a translational perspective, our findings affirm that BCL-2 immunohistochemical evaluation could be incorporated into standard MDS workups to help refine prognostic stratification and potentially identify those who might benefit most from BCL-2-targeted regimens. Of note, we found no correlation between BCL-2 levels and age, sex, or MDS subtype. These results align with the notion that MDS is molecularly heterogeneous and that, within a given morphologic category, disparate molecular drivers can dominate [22].

Although prior data have associated increased blasts with high BCL-2 expression, our study did not detect a direct link between blast percentage and H-score. This difference may reflect patient selection, our immunohistochemical methodology, or population size. Nonetheless, it does raise the possibility that high BCL-2 expression can pose a negative prognostic impact even when the blast percentage remains relatively low—an observation not uniformly emphasized in the existing literature [25,26].

Additionally, our patient population showed no strong correlation between BCL-2 status and infection rates or response to frontline therapies beyond hypomethylating agents. Some publications have proposed that BCL-2 positivity correlates with poor responses to conventional treatments [24,27], but our preliminary data do not confirm this. The relatively limited sample size and shorter follow-up interval in our cohort likely contributed to this difference, underscoring the need for larger, multi-institutional cohorts to validate such findings.

Because BCL-2 inhibitors (particularly venetoclax) have shown encouraging responses in MDS, future prospective trials could assess whether baseline or on-treatment BCL-2 immunostaining (or allied techniques such as flow cytometry or transcriptomics) can predict therapeutic success. Moreover, detailed subclonal analyses that dissect whether MCL-1, BCL-XL, or other anti-apoptotic factors eventually supplant BCL-2 might clarify how MDS cells escape monotherapy and which rational combination regimens (e.g., dual BCL-2/MCL-1 inhibitors, BCL-2 + FLT3 blockade, or BCL-2 + PD-1/PD-L1 immunotherapy approaches) could lead to more durable remissions [26,29].

Prospective studies that integrate quantitative BCL-2 immunohistochemistry with comprehensive mutational profiling upon diagnosis are warranted to clarify whether low-blast MDS cases with high BCL-2 expression harbor adverse driver mutations and to determine their potential sensitivity to BCL-2-directed therapy.

## Data Availability

The data that support the findings of this study are available from the corresponding author, Bartłomiej Kuszczak, upon reasonable request. The source data have been included in the Appendix A.

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
