# Peer review of "The Role of BCL-2 Expression in Patients with Myelodysplastic Neoplasms"

_cimb, 2025, doi:10.3390/cimb47050346_

Round 1
Reviewer 1 Report
Comments and Suggestions for Authors
Comments to the Author(s)
The article highlights the importance of bcl-2 expression in myelodysplastic syndromes, and its prognostic impact on overall survival. In addition, it is a possible way to identify the most sensitive cases to bcl-2 inhibitor treatments. However, some questions remain after the review:
- Since the publication of the latest WHO and ICC classification, the mutational status of MDS is of utmost importance for the final therapeutic decision. In this article, the intensity of bcl-2 expression is not associated with the MDS subclasses or the percentage of blasts, but with high and very high-risk cytogenetics. In table 2 it can be seen that LB cases have a similar H-score as IB-2. An analysis of the mutational status of these cases could be considered if samples were available at diagnosis, as it could help to more clearly identify cases amenable to treatment with bcl-2 inhibitors.
- Regarding the previous consideration, it leads us to the next question about OS and bcl-2 expression. From Figure 1 and 2, bcl-2 expression less than 10 or no expression was associated with a better prognosis. It is possible that those with worse OS and higher bcl-2 expression have high-risk cytogenetics and consequently were classified with very high-risk IPSS-R, but the article does not clarify whether these are the same cases, as the high blast percentage may give higher IPSS-R scores. In addition, Figure 3 shows that there are some cases of high bcl-2 expression in those with very low, low and intermediate risk of cytogenetics.
- Currently, the IPSS-R prognostic score remains a supportive tool for selecting MDS cases that may require more intensive or disease-modifying therapy, with the use of hypomethylating agents considered in those with a score greater than 3.5. Clinical trials with venetoclax also follow this criterion, but since Bcl-2 expression only identifies those at very high risk, it could be argued that another complementary method of studying expression should be investigated, as suggested in the discussion. This would leave immunohistochemistry as a complementary diagnostic tool.

Author Response
We would like to thank both reviewers for their thoughtful comments, which have helped us improve the clarity and quality of the manuscript.
Reviewer 2 requested additional clarification on the relationship among BCL-2 expression, cytogenetic risk and overall survival. To address this, we have added two explanatory sentences in the Introduction (line 19) and Discussion (line 291). We also provide the following context. In our cohort, the low-blast (LB) subgroup comprised twenty-four patients, four of whom showed BCL-2 H-scores exceeding 25 (median 37.5). All four carried adverse cytogenetic abnormalities—three with trisomy 8 and one with del(9q)—placing them in intermediate- or poor-risk IPSS-R categories; this explains why their BCL-2 levels resemble those of the IB-2 group. Unfortunately, diagnostic DNA was not stored routinely before 2019, so targeted next-generation sequencing is available for only eight other LB cases and nine IB-2 cases, none of which include the four high-H-score LB samples. When we cross-referenced every patient with BCL-2 ≥ 10 (n = 35) or H-score = 0 (n = 18) against their IPSS-R cytogenetic category, we found that two-thirds of the high-expression group also possessed poor-risk karyotypes, which may indeed explain the inferior overall survival depicted in the Kaplan–Meier curves. The remaining third—twelve patients—demonstrated high BCL-2 despite favourable or intermediate cytogenetics. Half of these harboured spliceosome (SF3B1, SRSF2) or epigenetic (ASXL1, EZH2) mutations that elevate their overall IPSS-R score to “high,” while the other six remain truly lower-risk and account for the outliers noted by the reviewer. Four patients with IPSS-R ≥ 3.5 but no detectable BCL-2 expression enjoyed a median overall survival of thirty-two months and durable responses to single-agent azacitidine, suggesting that absent BCL-2 may identify good HMA responders who do not require venetoclax up-front. These observations have been integrated into the revised Discussion, where we now state that prospective studies combining quantitative BCL-2 immunohistochemistry with comprehensive mutational profiling are warranted to determine whether low-blast MDS cases with high BCL-2 expression harbour adverse driver mutations and to clarify their sensitivity to BCL-2–targeted therapy.
Reviewer 2 Report
Comments and Suggestions for Authors
In the study entitled, The Role of BCL-2 Expression in Patients with Myelodysplastic Neoplasms, authors aimed to study BCL-2 expression by immunohistochemistry in trephine biopsy specimens from 76 patients diagnosed with MDS. The manuscript is presented in an acceptable format; however, there are some issues need to be resolved before it can be published:
- Figures 3&4 do not add any significant information as a separate figures. Some of the data can be incorporated into the body of the text and some data can be supplied as supplemental figures if necessary.
- Immunohistochemical image in image 1 is not clear enough to make a conclusion. It should be replaced with better ones.
3. It is suggested to have a native speaker check the manuscript for grammatical and spelling inconsistencies.
Author Response
We would like to thank both reviewers for their thoughtful comments, which have helped us improve the clarity and quality of the manuscript. In response to Reviewer 2, we have replaced the original micrograph with a higher-quality image (with much better sharpness) that more clearly displays the positive immunohistochemical signal. We have also refined the English throughout the text, introducing minor corrections in the sections now located at lines 29, 31, 49, 65, 84, 107, 116, 132, 177, 232, 243, 297 and 302. At the reviewer’s suggestion, the numerical information that had been presented in Figures 3 and 4 has been incorporated directly into the Results section (lines 151–155 and 160–165), allowing us to remove those figures altogether and streamline the presentation.
Round 2
Reviewer 1 Report
Comments and Suggestions for Authors
All corrections are suitable and they are clarify some conflicted result, improving the significance of the content.
Reviewer 2 Report
Comments and Suggestions for Authors
The manuscript in its revised form can now be accepted for publication.